# Spheno-Orbital Meningiomas: The Rationale behind the Decision-Making Process of Treatment Strategy

**DOI:** 10.3390/cancers16112148

**Published:** 2024-06-05

**Authors:** Giuseppe Mariniello, Sergio Corvino, Giuseppe Corazzelli, Oreste de Divitiis, Giancarlo Fusco, Adriana Iuliano, Diego Strianese, Francesco Briganti, Andrea Elefante

**Affiliations:** 1Department of Neurosciences, Reproductive and Odontostomatological Sciences, Division of Neurosurgery, School of Medicine, University of Naples “Federico II”, 80131 Naples, Italy; giuseppe.mariniello@unina.it (G.M.); sergio.corvino@unina.it (S.C.); giuseppe.corazzelli@studenti.unina.it (G.C.); oreste.dedivitiis@unina.it (O.d.D.); 2Department of Advanced Biomedical Sciences, School of Medicine, University of Naples “Federico II”, 80131 Naples, Italy; gianc.fusco@studenti.unina.it (G.F.); francesco.briganti@unina.it (F.B.); 3Department of Neurosciences, Reproductive and Odontostomatological Sciences, Division of Ophthalmology, School of Medicine, University of Naples “Federico II”, 80131 Naples, Italy; adriana.iuliano@unina.it (A.I.); strianes@unina.it (D.S.)

**Keywords:** spheno-orbital meningiomas, skull base meningiomas, sphenoid wing, cranio-orbital tumors, orbital tumors, endoscopic transorbital approach

## Abstract

**Simple Summary:**

“Surgery-first” is the main paradigm of treatment for intracranial meningiomas, with the aim of maximal safe tumor resection while preserving the neurological function. This purpose is not always achievable for meningiomas involving the spheno-orbital region, due to their close anatomical relationship with highly functional neurovascular structures, which limits the extent of resection. Therefore, surgery aims to achieve an onco-functional balance, mainly addressed to symptoms and signs of resolution. For this purpose, several surgical approaches, each with related pros and cons, can be considered.

**Abstract:**

Surgery stands as the primary treatment for spheno-orbital meningiomas, following a symptoms-oriented approach. We discussed the decision-making process behind surgical strategies through a review of medical records from 80 patients who underwent surgical resection at the University of Naples Federico II. Different surgical approaches were employed based on the tumor’s location relative to the optic nerve’s long axis, categorized into lateral (type I), medial (type II), and diffuse (type III). We examined clinical, neuroradiological, surgical, pathological, and outcome factors. Proptosis emerged as the most frequent symptom (97%), followed by visual impairment (59%) and ocular motility issues (35%). Type I represented 20%, type II 43%, and type III 17%. Growth primarily affected the optic canal (74%), superior orbital fissure (65%), anterior clinoid (60%), and orbital apex (59%). The resection outcomes varied, with Simpson grades I and II achieved in all type I cases, 67.5% of type II, and 18% of type III. Recurrence rates were highest in type II (41.8%) and type III (59%). Improvement was notable in proptosis (68%) and visual function (51%, predominantly type I). Surgery for spheno-orbital meningiomas should be tailored to each patient, considering individual characteristics and tumor features to improve quality of life by addressing primary symptoms like proptosis and visual deficits.

## 1. Introduction

Meningiomas are the most common primary central nervous system tumors, representing more than one third of all intracranial primary tumors [1,2]. Among them, spheno-orbital meningiomas (SOMs) account for 2–9% of all intracranial meningiomas [3] and mainly affect females (82%), who are usually younger than males at diagnosis, with a mean age of 51 ± 6 years old, and who more often express the progesterone receptor at histological examination [4]. Furthermore, the spheno-orbital region represents the most frequent location for intracranial meningiomas in the female sex [4].

In most cases, these tumors are slow-growing (0.3 cm^3^ per year) [5,6] and benign (WHO grade I). They arise from the arachnoid cap cells of the dura mater at the sphenoid wing with the secondary involvement of the orbit, commonly through the bone invasion of the lateral wall and roof of the orbit, or through the natural bony foramina represented by the superior orbital fissure (SOF) and/or the optic canal (OC), and they are characterized by an hyperostotic component and thin carpet-like soft tissue growth at the dura mater [7]. They represent a unique skull base tumor in terms of biological behavior and management [8].

Their peculiar pattern of growth accounts for the main presenting symptoms and signs, including proptosis, visual impairment, and ocular paresis [7]. Surgery represents the gold standard of treatment for symptomatic lesions, based on the paradigm of a “symptoms-oriented surgery”, and with the aim to restore/improve/arrest the progression of neuro-ophthalmological symptoms and signs. For this purpose, first, the understanding of physiopathology accounting for clinical manifestations, and which, in turn, guide the preoperative decision-making strategy of treatment, and later, the knowledge of both well-established and more recent surgical approaches for addressing these lesions are mandatory. In this setting, several surgical approaches have been developed over the years, both microsurgical and endoscopic, more or less invasive, each with associated pros and cons [9,10,11,12,13,14,15,16]; the selection depends on the goal of surgery and on patient and pathology features. The aim of the present study is to discuss our rationale behind the surgical strategy selection, discussing the main advantages and limits of the different approaches according to a topographic system targeting the optic nerve, through the analysis of clinical and surgical outcomes from a monoinstitutional surgical series.

## 2. Materials and Methods

### 2.1. Surgical Series

Medical record data of 80 patients with spheno-orbital meningiomas and who underwent surgery through microsurgical transcranial approaches at the University of Naples Federico II between 1990 and 2015 have been retrospectively reviewed. Inclusion criteria were the detection, using brain contrast-enhanced magnetic resonance imaging (MRI) and computed tomography (CT) scans, of both parenchymal and intraosseous tumor components and both intracranial and intraorbital compartment involvement, as well as tumor WHO grades I and II and patients with complete clinical, neuroradiological, surgical and outcome data.

### 2.2. Study Design

Demographic, clinical, neuroradiological, surgical, and outcome factors were analyzed. A careful clinical examination was performed by a neurosurgeon and an ophthalmologist; oculomotor nerves function, degree of proptosis, and visual acuity were evaluated. Neuroradiological study included the head CT scan to evaluate the invasion of the skull base bony structures, including the hyperostosis of the greater sphenoid wing, optic canal, superior orbital fissure, and anterior clinoid (AC); a brain contrast-enhanced MRI was used to define the parenchymal tumor component involving the intracranial dura mater and the periorbit, as well as the extradural tumor’s extension. Tumors were classified into 3 groups based on their anatomical relationship with an imaginary vertical plane passing along the long axis of the optic nerve on brain contrast-enhanced MRI, as follows: type I: lateral, type II: medial, type III: diffuse, when the optic nerve was affected in concentric manner. The microsurgical approach was selected according to the tumor type: lateral orbitotomy [10] for type I; supraorbital–pterional for types II and III. In addition, the medial orbital wall was removed in some tumors, type II and III, and the resection of the zygomatic arch was added to some tumors, type III. The extent of resection was evaluated through intraoperative assessment and/or on brain contrast-enhanced MRI at three months after surgery and classified according to the Simpson grading system [17], and it was considered to be complete for grades I and II and incomplete for grades III and IV. The clinical and surgical outcomes were assessed through clinical examination and seriated brain contrast-enhanced MRI and CT scan (at 1 and 2 months, at 1 year and every 2 years after surgery) during the follow-up (range from 5 to 28 years, median 136 months). An ANOVA Z-test was used for statistical analysis; *p* values < 0.05 were considered statistically significant.

## 3. Results

### 3.1. Demographic, Clinical, Neuroradiological, Surgical, and Outcome Data of the Overall Series (Table 1)

The overall sample was composed by 82.5% of females (n= 66/80) and 17.5% of males (n = 14/80), with a median age of 47 years (range 26–75 years). The main presenting clinical sign was proptosis, which was detected in all but one patient (n = 79/80, 97%). A deficit in visual acuity was reported in 47 cases (59%), while the disturbance of the ocular motility was reported in 28 cases (35%). According to the tumor pattern of growth relative to the long axis of the optic nerve, 20 (25%) were classified as type I—lateral, 43 (54%) as type II—medial, and 17 (21%) as type III—diffuse. Concerning the invasion of the skull base bony structures, the OC was the structure more often involved (n = 59/80, 74%), followed by the SOF (n = 52/80, 65%), ACP (n = 48/80, 60%), orbital apex (n = 47/80, 59%), ethmoid–sphenoid sinuses (n = 3/80, 4%), and infratemporal fossa (n = 3/80, 4%). Supraorbital–pterional was the most common approach adopted (n = 60/80, 75%), whereas lateral orbitotomy was reserved for the remaining cases (n = 20/80, 25%). The extent of resection, assessed according to the Simpson grading system [13], was considered to be complete (grade I and II) in most of the cases (n = 52/80, 65%) and incomplete in the remnants (n = 28/80, 35%). WHO grade I meningioma was diagnosed in 52 out of 80 patients (65%), whereas the remaining were WHO grade II tumors (n = 28/80, 35%). The clinical postoperative outcome was characterized by a significant improvement in proptosis in 68% of cases (n = 54/79), whereas it was stable or slightly improved in 32% (n = 25/79); the visual acuity significantly improved in 51% of cases (n = 24/47), whereas it was stable or slightly improved in 34% (n = 16/47) and worsened in the remaining 15% (n = 7/47). Finally, the deficit in ocular motility significantly improved in 43% of cases (n = 12/28), whereas it was stable or slightly improved in 39% (n = 11/28) and worsened in the remaining 18% (n = 5/28). Recurrences occurred in 30 cases (37.5%) during a mean follow-up of 136 months (range 5–28 years).

**Table 1 cancers-16-02148-t001:** Demographic, clinical, neuroradiological, surgical, and outcome data of the overall series of spheno-orbital meningiomas.

Covariates	N Cases 80 (%)
Age	Range 26–75 years (mean 47 y.o.)
Sex F M	66 (82.5%) 14 (17.5%)
Clinical Presentation Proptosis Visual acuity impairment Eye motility impairment	79 (97%) 47 (59%) 28 (35%)
Tumor Type I—Lateral II—Medial III—Diffuse	20 (25%) 43 (54%) 17 (21%)
Skull base bony structures invasion Optic canal Superior orbital fissure Anterior clinoid process Orbital apex Ethmoid–sphenoid sinuses Infratemporal fossa	59 (74%) 52 (65%) 48 (60%) 47 (59%) 3 (4%) 3 (4%)
Microsurgical Approach Supraorbital–pterional Lateral orbitotomy	60 (75%) 20 (25%)
Extent of resection (Simpson grade) I II III IV	21 (26%) 31 (39%) 21 (26%) 7 (9%)
WHO grade I II	52 (65%) 28 (35%)
Clinical Outcome • Proptosis: - Significant improvement; - Stable or minor improvement. • Visual acuity impairment: - Improved; - Stable; - Worsened. • Eye motility impairment: - Improved; - Stable; - Worsened.	54/79 (68%) 25/79 (32%) 24/47 (51%) 16/47 (34%) 7/47 (15%) 12/28 (43%) 11/28 (39%) 5/28 (18%)
Recurrence No Yes	50 (62.5%) 30 (37.5%)
Follow-up	Range 5–28 years (Mean 136 months)

### 3.2. Skull Base Invasion and Surgical and Pathological Findings according to Tumor Type (Table 2 and Table 3)

The optic canal was the only skull base bony structure involved in type I tumors (n = 3/20, 15%). Conversely, the orbital apex, the superior orbital fissure, and the optic canal were always involved in type III tumors (n = 17/17, 100%) and in most (70–90% of cases) type II tumors. The anterior clinoid was affected with a similar incidence rate in type II (79%) and type III (82%) tumors. The ethmoid–sphenoid sinuses were invaded in two cases belonging to type II (5%) and one case among type I (6%) tumors. Finally, the infratemporal fossa was only involved in type III tumors (n = 3/17, 18%).

All these data are summarized in Table 2.

Twenty cases belonging to type I tumors (20/80, 25%) were treated through lateral orbitotomy, and a complete tumor resection (Simpson grade I 65% and II 35%) was achieved in all cases. The remaining 60 cases (75%), including type II and III tumors, were treated by the supraorbital–pterional approach, with the adjunct of medial orbital wall removal and medial decompression of the optic canal in two cases of type II tumors and one case of type III tumor for tumor extension into the ethmoid–sphenoid sinuses. In the three cases (n = 3/17, 18%) of infratemporal fossa tumor extension belonging to type III tumors, the supraorbital–pterional was converted into a fronto-temporo-orbito-zygomatic approach. The supraorbital–pterional approach allowed for the achievement of complete tumor resection in 68% of type II tumors (Simpson grade I 19% and II 49%) and 18% of type III tumors (Simpson grade II); vice versa, incomplete resection resulted in 32% of type II tumors (n = 14/43, Simpson grade III 14/14) and 82% of type III tumors (n = 14/17, Simpson grade III 7/14 and Simpson grade IV 7/14). WHO grade I was diagnosed in 75% of type I tumors (n = 15/20), 65% of type II tumors (n = 28/43), and 53% of type III tumors (n = 9/17).

All these data are summarized in Table 3.

**Table 2 cancers-16-02148-t002:** Skull base invasion according to tumor type.

Tumor Location	N Patients (%)	Orbital Apex	Optic Canal	Superior Orbital Fissure	Anterior Clinoid	Ethmoid- Sphenoid Sinus	Infra-Temporal Fossa
Type 1 Lateral	20 (25%)	—	3 (15%)	—	—	—	—
Type II Medial	43 (54%)	30 (70%)	39 (90%)	35 (81%)	34 (79%)	2 (5%)	—
Type III Diffuse	17 (21%)	17 (100%)	17 (100%)	17 (100%)	14 (82%)	1 (6%)	3 (18%)
Total	80 (100%)	47 (59%)	59 (74%)	52 (65%)	48 (60%)	3 (4%)	3 (4%)
Statistic		Type I vs. II *p* < 0.01 Type II vs. III *p* < 0.005 Type I vs. III *p* < 0.005	Type I vs. II *p* < 0.002 Type II vs. III *p* = n.s. Type I vs. III *p* < 0.001	Type I vs. II *p* < 0.008 Type II vs. III *p* = 0.02 Type I vs. III *p* < 0.005	Type I vs. II *p* < 0.002 Type II vs. III *p* = n.s. Type I vs. III *p* < 0.001	Type I vs. II *p*= n.s. Type II vs. III *p* = n.s. Type I vs. III *p* = n.s.	Type I vs. II n.s. Type II vs. III *p* = 0.002 Type I vs. III *p* = 0.023

n.s.: not significant.

**Table 3 cancers-16-02148-t003:** Surgical and pathological findings according to tumor type.

Tumor Location/ Num.	Simpson Grade Resection				WHO Grade	
	I	II	III	IV	I	II
Type 1 Lateral (20)	13 (65%)	7 (35%)	—	—	15 (75%)	5 (25%)
Type II Medial (43)	8 (19%)	21 (49)	14 (32%)	—	28 (65%)	15 (35%)
Type III Diffuse (17)	—	3 (18%)	7 (41%)	7 (41%)	9 (53%)	8 (47%)
Total (80)	21 (26%)	31 (39%)	21 (26%)	7 (9%)	52 (65%)	28 (35%)
Statistic	*p* = n.s.				*p* = n.s.	

n.s.: not significant.

### 3.3. Clinical Outcome According to Tumor Type (Table 4)

Proptosis improved after surgery in most patients (n = 54/79, 68%), whereas it remained stable in the remnants (n = 25/79, 32%). According to the location, type I tumors were associated with a significantly lower rate of visual disfunction (n = 3/17, 15%) compared to type II (n = 32/43, 74%) and type III tumors (n = 12/17, 71%) (Table 3). The optic canal was involved in all cases of type III tumors (n = 17/17, 100%), in 90% among type II tumors (n = 39/43), and in 15% of type I tumors (n = 3/20). Visual outcome registered a complete resolution or a significant improvement in the preoperative visual deficit in 100% of type I tumors (n = 3/3), in 53% of type II tumors (n = 17/32), and 33.3% of type III tumors (n = 4/12). On the other hand, only seven (n = 7/47, 15%) cases referring to a postoperative worsening of visual function belonged to type II (n = 4/7) and III (3/7) tumors.

All these data are summarized in Table 4.

**Table 4 cancers-16-02148-t004:** Clinical outcome according to tumor type.

Tumor Type	Preoperative Visual Dysfunction	Optic Canal Involvement	Visual Outcome		
			Remission or Significant Improvement	Slight Improvement or Stable	Worsening
Type 1 Lateral (20)	3 (15%)	3 (15%)	3 (100%)	—	—
Type II Medial (43)	32 (74%)	39 (90%)	17 (53%)	11 (25.5%)	4 (9%)
Type III Diffuse (17)	12 (71%)	17 (100%)	4 (33.3%)	5 (41.7%)	3 (25%)
Total (80)	47 (59%)	59 (74%)	24 (51%)	16 (34%)	7 (15%)
Statistic	Type I vs. II *p* < 0.005 Type II vs. III *p* = n.s. Type I vs. III *p* < 0.002	Type I vs. II *p* < 0.002 Type II vs. III *p* = n.s. Type I vs. III *p* < 0.001	Type I vs. II *p* = 0.021 Type II vs. III *p* = n.s. Type I vs. III *p* = 0.019	Type I vs. II *p* = n.s. Type II vs. III *p* = n.s. Type I vs. III *p* = n.s.	Type I vs. II *p* = n.s. Type II vs. III *p* = n.s. Type I vs. III *p* = n.s.

n.s.: not significant.

## 4. Discussion

### 4.1. From Pathology to Clinics

The natural history of spheno-orbital meningiomas is characterized by the “rule of two”: two anatomical compartments affected (intracranial and orbit), two tissue components (parenchymatous and osseous), and two patterns of growth (round and flat). SOMs arise from the dura covering the sphenoid wing with secondary extension to the orbit and surrounding neurovascular structures via the bony invasion of the lateral wall and/or roof of the orbit and of the soft tissues. They consist of a parenchymatous tumoral tissue involving the temporo-polar dura mater and periorbit and a hyperostosis of various degrees involving the bony tissue of the orbit, especially its lateral wall, roof, and optic canal. Finally, the parenchymatous tumor can exhibit two patterns of growth: en-plaque (or flat), a thin carpet-like soft tissue growth at the dura mater, and globous (or round) with intradural growth. The hard consistency of the hyperostotic bone tissue of the lateral wall, as well as of the roof of the orbit, is the main cause for proptosis [5,18,19], which represents the main presenting symptoms of SOMs [5,9,10,14,18,19,20], and which usually is unilateral, non-pulsating, irreducible, and slow evolving. Despite some authors also hypothesizing periorbital tumor invasion, intraorbital tumor, and venous stasis from the compression of the ophthalmic vein as other causes of proptosis [21], the hyperostosis–proptosis correlation as a cause–effect is further supported by the significant correlation between intraosseous tumor volume resected and proptosis improvement, with success rates ranging from 50 to 100% [9,19,22,23,24]. The involvement of the optic canal by hyperostosis, as well as by parenchymatous tumor invasion, accounts for the second most common presenting symptom of SOMs, i.e., visual acuity deficit [9,23,25,26]. Also, in this case, this relationship is widely demonstrated by the arrest of deterioration or improvement in visual acuity after unroofing the optic canal and releasing visual neurovascular structures [25,27,28,29]. In addition, it is worthy of remembering that optic canal decompression is the most frequent surgical maneuver performed (82%) in the surgery of spheno-orbital meningiomas and the improvement in visual acuity and visual field deficits occur in 91% and 87% of cases, respectively [9].

### 4.2. From Clinics to Surgery: Modular Approaches

Spheno-orbital meningiomas represent a unique skull base tumor in terms of natural history as well as of management. Guidelines of treatment exist for intracranial meningiomas and recommend maximal safe tumor resection while preserving neurological functions for symptomatic lesions in good-performance-status patients [30]. This purpose is not always achievable for SOMs due to their critical location of being close to highly functional neurovascular structures. Therefore, there is not unanimous consent regarding their management, which often varies among single institutions, with centers adopting the strategy of subtotal resection followed by radiation therapy or by “wait and see” and multiple re-operations over the years when symptoms occur [16,20,26,31,32,33,34,35,36,37,38,39,40,41,42,43]. Over the years, thanks to better knowledge being available on the natural history of this tumor and the refinements of the fields of surgery, radiation therapy, and technology, there has been a progressive “change in paradigm” of treatment, switching from the main goal of surgery of a gross total tumor resection to ensure the best functional outcome for the patient, with a special focus on preserving/restoring visual function and correct proptosis through orbital decompression while attempting to achieve maximum safe tumor resection with minimal postoperative complications. In this scenario, the surgical techniques addressing SOMs have also evolved from aggressive approaches involving extensive craniotomies, with the most adopted represented by fronto-temporal [9], through less invasive lateral orbitotomy [10,44], to the more recent endoscopic endonasal [45,46] and transorbital [13,15,47,48] techniques, each with related pros and cons, but without a well-defined indication for their selection. In the past, our group proposed a surgical strategy of treatment through different microsurgical transcranial approaches based on the topographic pattern of the lesion relative to the long axis of the optic nerve [14,31]. The advent of the endoscopic approaches in the neurosurgical practice, via the endonasal technique first, and more recently via the transorbital approach, has expanded the routes used to access these lesions [12,45,46,48,49,50,51,52,53,54,55,56]. Recently, Kong et al. [13] provided an anatomical classification from a surgical endoscopic transorbital perspective based on the location of the tumor epicenter on the greater sphenoid wing, identifying three types by dividing that region into three thirds: medial, middle, and lateral. Later, Baucher et al. [57], starting from the morphological classification by Roser et al. [58] and the anatomical classification of Kong et al. [13], added the parameter of tumor invasion of seven specific anatomical regions and structures: temporal fossa, infratemporal fossa, orbit, superior orbital fissure, anterior clinoid process, optic canal, and cavernous sinus. Therefore, when optic canal decompression is the main goal together with maximal safe tumor resection, a schematic approach to surgical strategy selection can be considered, focusing on the mantra that the pattern of the growth of the lesion around the optic canal drives the selection of the approach. Several anatomical quantitative cadaveric studies have compared optic nerve decompression via transcranial, endoscopic endonasal, and endoscopic transorbital approaches in terms of circumferential and longitudinal bone removal [50,51,59]; some authors observed that the largest circumferential decompression was provided by the extended pterional approach (mean 245°), which addressed the superolateral 68% of the optic canal circumference, followed by the transorbital corridor (mean from 178° to 192°), which accounted for 50–53.3% of supero-lateral optic canal decompression and the endonasal route (mean from 145° to 168°), which addressed the inferomedial 40–46.7% optic canal decompression [50,51]. In terms of the length of OC decompression, open transcranial and endoscopic endonasal provided similar results (mean length 13 mm) [59]. These findings were matched in clinical practice [10,14,27,46,48,60]. For tumors limited to the lateral (type I) or superolateral compartments of the orbit, lateral orbitotomy [10] can be considered, allowing for the lateral decompression of the OC. However, invasion of the optic canal and tumor extension beyond the optic nerve are contraindications to the approach via lateral orbitotomy. For tumors involving the superomedial aspect (type II), or in a diffuse and concentric manner the optic canal (type III), the open microsurgical transcranial approach, as such as supraorbital–pterional, can be considered. In these cases, this approach allows for an extensive uncovering of the ON, except in its inferomedial aspect. This anatomical–surgical limit can be overcome by combining an endoscopic endonasal approach. Endoscopic endonasal approaches (EEAs) provide extraordinary ventral surgical corridors to access the selected pathologies of the midline skull base [61,62]. This corridor, after the removal of the lamina papyracea, allows for the ON to be exposed from the Zinn annulus to the entry on the OC, as well as the unroofing of the medial wall of the OC up to the lateral edge of the tuberculum sellae. However, in a recent literature review on the treatment of spheno-orbital meningiomas via the EEA [45], including eight studies and 19 cases, this approach was adopted in an isolated manner just once (5.3%), while it was adopted in a combined one in the remaining 18 cases (94.7%). The endoscopic endonasal technique affords great visualization of the orbital apex and optic canal and allows for the resection of tumors extended medially to the optic canal, pterygopalatine fossa, and the infratemporal fossa. The endoscopic transorbital approach (ETOA), first mainly adopted by ophthalmologists, allows for lesions affecting the paramedian aspect of the anterior and middle cranial fossae to be accessed [47,63,64,65,66,67]. This route addresses a similar anatomical target of the OC to the microsurgical transcranial approach, including lateral orbitotomy, but with different angles of attack, surgical freedom, and carrying peculiar benefits and limits [15]. In a recent literature review [16], open craniotomies and type I spheno-orbital meningiomas were found to be associated with the highest rate of gross total resection, whereas the ETOA, either as an isolated or combined approach with the EEA, provides the lowest rate of gross total resection. Conversely, the ETOA and type I SOMs are associated with the highest rates of postoperative visual acuity and proptosis improvement. These data can be explained by the wider exposure and working areas, as well as of surgical freedom, related to transcranial approaches and the complete removal of the lateral orbital wall, including the hyperostotic bone, accounting for proptosis via the ETOA. Because of the lower rates of gross total resection, endoscopic approaches should be reserved for selected patients with hypotheses of low-grade tumors, a small parenchymal component, and prevalent hyperostosis. The main postoperative complications include a cranial nerve deficit for transcranial approaches and CSF leak for endoscopic endonasal and transorbital corridors. As SOMs can involve multiple compartments of the orbit and different sides of the optic nerve and canal, even more than one approach can be considered, performed in an isolated or combined manner, in a single stage or multiple stages. To conclude, the goal of surgery in spheno-orbital meningiomas is to achieve onco-functional balance; for this purpose, the choice of an aggressive surgical approach might lead to unnecessary peri- and postoperative morbidity. On the other hand, a less invasive and more conservative approach might not provide adequate exposure of the surgical target area, not guarantee the control of neurovascular structures, a satisfying bony decompression, and tumor removal, resulting in no clinical improvement and a high rate of recurrence. The treatment must be tailored for each patient.

## 5. Conclusions

Different approaches, each with related pros and cons, are in the armamentarium of the neurosurgeon to address spheno-orbital meningiomas. Optic nerve decompression in the optic canal represents one of the main goals of surgery when preoperative visual deficit occurs. Different surgical corridors can be adopted in isolated or variously combined manners according to the goal of surgery and the morphological involvement of the optic nerve by pathology. Their combination allows for a 360 degrees decompression of the optic nerve.

## Data Availability

The data presented in this study are available on request from the corresponding author due to privacy restrictions.

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
