# Peer review of "Spheno-Orbital Meningiomas: The Rationale behind the Decision-Making Process of Treatment Strategy"

_cancers, 2024, doi:10.3390/cancers16112148_

Round 1
Reviewer 1 Report
Comments and Suggestions for Authors
Dear Editor,
I have reviewed the paper entitled "Spheno-Orbital Meningiomas: The Rationale Behind the Decision-Making Process of Treatment Strategy" by Mariniello et al.
They had reviewed 80 patients that underwent surgeries through microsurgical transcranial approaches
due to Spheno-Orbital Meningiomas and evaulated many parameters regarding surgical approach, tumor characteristics and outcome.
The definition of "symptoms-oriented approach" is very useful in order to achieve oncofunctional balance.
However, the tables are difficult to understand or evaluate.
I suggest the authors to compare the mentioned different surgical approaches for such pathology not only to review the microscopic approach that I am not sure in current form what adds to current literature.
Best regards
Comments on the Quality of English LanguageDear Editor,
I have reviewed the paper entitled "Spheno-Orbital Meningiomas: The Rationale Behind the Decision-Making Process of Treatment Strategy" by Mariniello et al.
They had reviewed 80 patients that underwent surgeries through microsurgical transcranial approaches
due to Spheno-Orbital Meningiomas and evaulated many parameters regarding surgical approach, tumor characteristics and outcome.
The definition of "symptoms-oriented approach" is very useful in order to achieve oncofunctional balance.
However, the tables are difficult to understand or evaluate.
I suggest the authors to compare the mentioned different surgical approaches for such pathology not only to review the microscopic approach that I am not sure in current form what adds to current literature.
Best regards
Author Response
I have reviewed the paper entitled "Spheno-Orbital Meningiomas: The Rationale Behind the Decision-Making Process of Treatment Strategy" by Mariniello et al.
They had reviewed 80 patients that underwent surgeries through microsurgical transcranial approaches due to Spheno-Orbital Meningiomas and evaulated many parameters regarding surgical approach, tumor characteristics and outcome.
The definition of "symptoms-oriented approach" is very useful in order to achieve oncofunctional balance.
However, the tables are difficult to understand or evaluate.
I suggest the authors to compare the mentioned different surgical approaches for such pathology not only to review the microscopic approach that I am not sure in current form what adds to current literature
Answer: Dear Reviewer, thank you for your time, for your comments and for appreciating our manuscript.
We agree with you that the Tables could be complicated for an immediate evaluation; therefore, as you suggested, results summarized in the Tables have been simplified for a better and more immediate understanding. We separated the tables according to the analyzed factors and reported in bold only the results statistically significant.
Concerning the proposal of comparison of the different approaches, we extensively discussed the role of the different surgical approaches with respect to the goal of surgery in spheno-orbital meningiomas, which is the topic of our study. We did not perform a proper review as it was beyond the aim of our study. We agree with you that a comparison between the three considered corridors (transcranial, endoscopic endonasal and transorbital) focused on other aspects is interesting, however, several literature reviews on this topic have been recently proposed and cited in the Discussion section of the text.
Because of the long experience of our group in the scientific research and in the microsurgical management of intracranial meningiomas, especially the spheno-orbital ones, we preferred focusing on the rationale behind the surgical strategy selection in the light of the more or less recent endoscopic approaches, considering an anatomical-surgical classification that we proposed in the past and discussing the main advantages and limits of the different surgical corridors.
Reviewer 2 Report
Comments and Suggestions for Authors
Dear colleagues!
Thank you for the interesting research, which is presented at a good methodological level. In general, I have no special comments, but it is necessary to arrange the tables according to the standard, and also write a more meaningful introduction, covering, for example, epidemiology and the prevalence of pathology.
In this regard, it will be necessary to update and update the list of references
Author Response
Thank you for the interesting research, which is presented at a good methodological level. In general, I have no special comments, but it is necessary to arrange the tables according to the standard, and also write a more meaningful introduction, covering, for example, epidemiology and the prevalence of pathology.
Answer: Dear Reviewer, thank you so much for your time, your suggestions and for appreciating our study. We agree with your comments, therefore the “Introduction” section has been enriched and the tables have been arranged according to the standard. The following sentences have been added:
“Meningiomas are the most common primary central nervous system tumors, rep-resenting more than one third of all intracranial primary tumors[1,2]. Among them, spheno-orbital meningiomas (SOMs) account for 2-9% of all intracranial meningiomas[3], mainly affect females (82%), who usually are younger than males at diagnosis, with a mean age of 51±6 years old and who more often express the progesterone receptor at histological examination[4]. Furthermore, the spheno-orbital region represents the most frequent location for intracranial meningiomas in sex female[4].
In most cases these tumors are slow-growing (0,3 cm3 per year)[5,6] and benign (WHO grade I).”
Reviewer 3 Report
Comments and Suggestions for Authors
This study presented a retrospective analysis of spheno-orbital meningiomas in the period of 25 years. The follow-up period is also long (at least 5 years). This enabled authors to evaluate a large number of patients over a long period of time. The result is a comprehensive analysis focusing on the anatomical location (lateral, medial, diffuse), type of the approach and final results.
The method and result section are described adequately, providing an honest view on the outcome. In the discussion section, the alternative approaches (including endonasal) are described, especially for optic nerve decompression. The study contributes important information on the treatment of these rare entities.
Overall, it is a well-designed study and I propose to be published in the current form.
Author Response
Reviewer #3
This study presented a retrospective analysis of spheno-orbital meningiomas in the period of 25 years. The follow-up period is also long (at least 5 years). This enabled authors to evaluate a large number of patients over a long period of time. The result is a comprehensive analysis focusing on the anatomical location (lateral, medial, diffuse), type of the approach and final results.
The method and result section are described adequately, providing an honest view on the outcome. In the discussion section, the alternative approaches (including endonasal) are described, especially for optic nerve decompression. The study contributes important information on the treatment of these rare entities.
Overall, it is a well-designed study and I propose to be published in the current form.
Answer: Dear Reviewer, we would like to express our sincere gratitude for your comments and your time; we are proud that you appreciated our study, and we agree with your suggestions.